# Estimating time-varying cholera transmission and oral cholera vaccine effectiveness in Haiti and Cameroon, 2021-2023

Erin N. Hulland[1,2], Marie-Laure Charpignon[2,3], Ghinwa Y. El Hayek[2,4], Lihong Zhao[2,5], Angel N. Desai[2,6], Maimuna S. Majumder[2,7]

## Abstract

In 2023, cholera affected approximately 1 million people and caused more than 5000 deaths globally, predominantly in low-income and conflict settings. In recent years, the number of new cholera outbreaks has grown rapidly. Further, ongoing cholera outbreaks have been exacerbated by conflict, climate change, and poor infrastructure, resulting in prolonged crises. As a result, the demand for treatment and intervention is quickly outpacing existing resource availability. Prior to improved water and sanitation systems, cholera, a disease primarily transmitted via contaminated water sources, also routinely ravaged high-income countries. Crumbling infrastructure and climate change are now putting new locations at risk — even in high-income countries. Thus, understanding the transmission and prevention of cholera is critical.

Combating cholera requires multiple interventions, the two most common being behavioral education and water treatment. Two-dose oral cholera vaccination (OCV) is often used as a complement to these interventions. Due to limited supply, countries have recently switched to single-dose vaccines (OCV1). One challenge lies in understanding where to allocate OCV1 in a timely manner, especially in settings lacking well-resourced public health surveillance systems. As cholera occurs and propagates in such locations, timely, accurate, and openly accessible outbreak data are typically inaccessible for disease modeling and subsequent decision-making.

In this study, we demonstrated the value of open-access data to rapidly estimate cholera transmission and vaccine effectiveness. Specifically, we obtained non-machine readable (NMR) epidemic curves for recent cholera outbreaks in two countries, Haiti and Cameroon, from figures published in situation and disease outbreak news reports. We used computational digitization techniques to derive weekly counts of cholera cases, resulting in nominal differences when compared against the reported cumulative case counts (i.e., a relative error rate of 5.67% in Haiti and 0.54% in Cameroon). Given these digitized time series, we leveraged EpiEstim—an open-source modeling platform—to derive rapid estimates of time-varying disease transmission via the effective reproduction number ($R_t$). To compare OCV1 effectiveness in the two considered countries, we additionally used VaxEstim, a recent extension of EpiEstim that facilitates the estimation of vaccine effectiveness via the relation among three inputs: the basic reproduction number ($R_0$), $R_t$, and vaccine coverage. Here, with Haiti and Cameroon as case studies, we demonstrated the first implementation of VaxEstim in low-resource settings. Importantly, we are the first to use VaxEstim with digitized data rather than traditional epidemic surveillance data.

In the initial phase of the outbreak, weekly rolling average estimates of $R_t$ were elevated in both countries: 2.60 in Haiti [95% credible interval: 2.42-2.79] and 1.90 in Cameroon [1.14-2.95]. These values are largely consistent with previous estimates of $R_0$ in Haiti, where average values have ranged from 1.06 to 3.72, and in Cameroon, where average values have ranged from 1.10 to 3.50. In both Haiti and Cameroon, this initial period of high transmission preceded a longer period during which $R_t$ oscillated around the critical threshold of 1. Our results derived from VaxEstim suggest that Haiti had higher OCV1 effectiveness than Cameroon (75.32% effective [54.00-86.39%] vs. 54.88% [18.94-84.90%]). These estimates of OCV1 effectiveness are generally aligned with those derived from field studies conducted in other countries. Thus, our case study reinforces the validity of VaxEstim as an alternative to costly, time-consuming field studies of OCV1 effectiveness. Indeed, prior work in South Sudan, Bangladesh, and the Democratic Republic of the Congo reported OCV1 effectiveness ranging from approximately 40% to 80%.

This work underscores the value of combining NMR sources of outbreak case data with computational techniques and the utility of VaxEstim for rapid, inexpensive estimation of vaccine effectiveness in data-poor outbreak settings.

**Keywords:** Cholera, Haiti, Cameroon, reproduction number, $R_0$, Effective R, statistical modeling, computational epidemiology, outbreak and response, vaccination, infectious disease, low-income countries, OCV, data scarcity, vaccine effectiveness

## Introduction

Cholera, a disease characterized by severe watery diarrhea caused by toxigenic serogroups O1 or O139 of the bacteria *Vibrio cholerae*, has ravaged human society through seven deadly pandemics since its first documented international spread in 1817.[1] The seventh pandemic, which began in 1961, is still ongoing. Presently, cholera affects about one quarter of the world's countries and hundreds of millions of people. Although the disease predominantly impacts low-income countries and conflict settings,[1,2] most developing countries also remain at risk for an outbreak.[3,4] More alarmingly, since 2022, an increasing number of countries have been reporting new cholera outbreaks. Moreover, due to limited resources, worsening political instability and human displacement, and poor infrastructure, most of these outbreaks are continuing.[5]

Such crises have been widespread throughout the globe, from Haiti in the Caribbean, to Democratic Republic of Congo, Malawi, and Cameroon in Africa, to Lebanon and Afghanistan in the Middle East, to India and China in Asia.[6]

Cholera spreads through contaminated water either by naturally occurring bacteria or through fecal transmission in water by infected individuals. Cholera is of particular concern in locations with limited access to improved water and sanitation systems, as outbreaks persist and propagate where water access, sanitation, and hygiene systems are insufficient.[7] Although improving these systems is the most effective way to prevent onward cholera transmission in the long run, infrastructure development is costly and time-intensive, and therefore unsuitable as a primary intervention during an ongoing outbreak.[8,9] In the short term, treatment of mild to moderate cases is instead often well-managed with oral rehydration therapy (ORT), a low-cost and rapid intervention. However, both access to and adoption of ORT remain limited in many low-income countries. As a result, cholera is still a significant contributor to global morbidity and mortality, with an estimated 1.2 to 4 million cases and an average of 95,000 deaths annually.[10–14] While high-income countries have been spared in recent decades, they are not entirely immune to the threat cholera represents. In the United States, for example, millions of people lack clean water or complete indoor plumbing. Additionally, most of the water and sanitation infrastructure in the country is outdated and requires immediate replacement or repair to maintain satisfactory performance.[15–19] Moreover, climate change, which influences water quality, quantity, and access via drought, flooding, and other catastrophic weather events, poses a threat for many existing water and sanitation systems globally.[20] Thus, while the burden of cholera is currently concentrated in low-income and conflict settings, high-income countries are far from immune to a resurgence of cholera and other waterborne or zoonotic threats. Hence it is important for epidemiologists and mathematical modelers to consider the additional risks posed by climate change for disease resurgence, both in the US and abroad.[21]

In outbreak contexts, vaccination with killed whole-cell oral cholera vaccines (OCV) is considered a safe and effective intervention when cases are surging. Typically, vaccination with OCVs generally requires administration of two doses. Three main OCVs are recommended by the World Health Organization for cholera intervention — Shanchol, Euvichol, or Dukoral — but only Shanchol and Euvichol are available for mass vaccination campaigns through the Gavi-supported global stockpile.[22] Both Shanchol and Euvichol have generally shown reasonable to high effectiveness: field studies in Haiti, Bangladesh, Guinea, Zambia, and India demonstrated that two-year effectiveness of two-dose vaccines ranged from 37% to 97.5%, averaging around 65%.[23–28] This variability in vaccine *effectiveness*, rather than *efficacy*, reflects in part country-level differences in the strength of health systems and challenges associated with vaccine distribution in urban vs. rural areas. However, the global stockpile of OCV — established in 2013 — has recently faced shortages, with demand consistently outpacing supply despite annual increases in procurement.[29,30] As of October 2022, the International Coordinating Group for vaccine delivery has thus restricted outbreak vaccination campaigns to only single-dose vaccines (OCV1).[31] Studies recently conducted in South Sudan, Bangladesh, and the Democratic Republic of the Congo suggest that the short-term effectiveness of OCV1 ranges between 40% and 80% across settings, but that it is lower over longer time periods.[32–34] In most countries where both OCV2 and OCV1 campaigns were conducted, such short-term estimates of single-dose effectiveness largely align with their two-dose counterparts, adding evidentiary support to the use of OCV1 when facing shortages.

Using Haiti and Cameroon as compelling examples of the 20+ countries experiencing new or worsening outbreaks in 2022,[6] we sought to investigate cholera transmission dynamics, evaluate vaccine effectiveness, and understand the role of sociopolitical and health systems as well as climate. OCV1 campaigns were conducted to aid in reducing cholera transmission in both countries. As such, they can serve as interesting natural case studies to assess one- versus two-dose vaccine effectiveness and identify location-specific variations. While Haiti and Cameroon share risk factors exacerbating their vulnerability to cholera, their local climate, health systems, and existing water and sanitation infrastructure differ. Such heterogeneity allows us to interrogate associations between these factors and vaccine effectiveness and to formulate hypotheses regarding their potential impact during subsequent vaccination campaigns in the two countries, which may be generalizable to other locations. By helping us understand how the patterns of cholera transmission and vaccine effectiveness vary by context, this case study can inform future vaccination efforts worldwide.

The outbreak currently affecting Cameroon began in 2021, while Haiti's first case was not reported until 2022. To this date, disease transmission persists in both countries, despite numerous ebbs and flows in reporting.[4,35–38] OCV1 campaigns occurred in February and April 2022 in Cameroon and in December 2022 in Haiti. While Cameroon has had recurrent outbreaks since cholera was first reported in the country in 1971,[39,40] Haiti had never had a single confirmed cholera case until 2010, in the aftermath of the deadly earthquake in Port-au-Prince. While the basic and effective reproduction numbers ($R_0$ and $R_t$) – two related but distinct measures of transmissibility – have been used as the key metrics to understand cholera outbreaks, most research has employed mechanistic compartmental models requiring data availability, mathematical modeling expertise, and assumptions about the underlying disease process and corresponding parameterization. Because these are poorly suited to outbreak conditions due to limitations in data availability and computing resources, we explored an

alternative approach, capitalizing on phenomenological models which do not rely on assumptions about the disease process and are generally more parsimonious.

A distinct feature of our approach was to start with the digitization of case count data from outbreak and situation reports. Unlike in high-income settings, timely, open-access, machine-readable surveillance data of infectious disease case counts are frequently inaccessible for modeling; thus, the use of alternative data sources is paramount. Upon digitizing published epidemic curves, we obtained the trajectory of cholera case counts in each country. Next, we used the openly available R package EpiEstim[41–43] to estimate the time-varying effective reproduction number $R_t$. This step requires passing four epidemiological parameters, all of which can be gleaned from published literature. Last, we extended our contribution to also estimate vaccine effectiveness in each country using the canonical relation between $R_t$, $R_0$, vaccine coverage, and vaccine effectiveness.[44] This step was facilitated by a recent extension of the EpiEstim package called VaxEstim. So far, this extended framework has only been applied within the United States and to readily accessible, high-quality data; this work provides its first illustration in a low-resource setting using digitized case count data.[45]

## Methods

### Basic and effective reproduction numbers

In epidemiology, the basic reproduction number ($R_0$) measures the average number of secondary infections caused by a primary infection over the course of the infectious period, assuming that the entire population is susceptible.[46] When the population is not fully susceptible, due to either vaccination or natural immunity, the effective reproduction number ($R_t$) is used instead of $R_0$. Using available epidemic curves, $R_t$ can be readily estimated at each time point $t$ via the Bayesian framework proposed by Cori et al. and by leveraging the companion R package, EpiEstim.[41–43] When data about vaccine coverage are available, estimates of vaccine effectiveness can be derived from the above estimates of $R_t$, along with $R_0$ values obtained from the literature. In this study, we built upon previous uses of EpiEstim[47–54] and applied VaxEstim[45] to model cholera outbreaks in two distinct countries, Haiti and Cameroon.

### Data sourcing and cleaning

Without readily-accessible, machine-readable cholera surveillance data, we capitalized on published epidemic curves from situation reports and disease outbreak news reports. Although estimates of OCV1 coverage were reported infrequently in situation and disease outbreak news reports, we extracted every mention for subsequent use in vaccine effectiveness models. For Haiti, we leveraged situation reports released by the Pan-American Health Organization (PAHO) and the Ministère de la Santé Publique et de la Population (MSPP).[35,38] For Cameroon, we used disease outbreak news reports from the WHO.[4]

In Cameroon, the most recent outbreak of cholera was declared on October 29, 2021. By May 2022 – the last published disease outbreak news report at the time of analysis – over 6,000 confirmed or suspected cases and 370 deaths had been registered.[55] In Haiti, the most recent outbreak was confirmed on October 1, 2022, following a three-year hiatus without any cholera cases. By August 2023 – the last report published by PAHO at the time of analysis – over 58,000 confirmed or suspected cases and 800 deaths had been attributed to this outbreak.[35,37,38]

Information about the serial interval (SI) — the time elapsed between the onset of symptoms in two consecutive cases — was obtained from previous modeling studies on cholera,[50,52] as well as from prospective and retrospective epidemic assessments.[56,57] Based on previous studies, we assumed that the SI followed a Gamma distribution with a mean of 4 days and a standard deviation of 3.[50,52] Data on the historic $R_0$ for cholera were gleaned from national and regional reports of prior outbreaks and other modeling studies. Given the noted importance of using country-specific or regional data to estimate $R_0$, we used different priors for Haiti and for Cameroon. In our main analysis, we considered a mean $R_0$ of 2.27 for Haiti, estimated using previous country-specific publications.[58–62] We considered a mean $R_0$ of 1.95 for Cameroon, estimated using one country-specific study[63] and two regional analyses.[50,52] In Appendix Figure 5, we separately investigated the local sensitivity of our vaccine effectiveness results to the choice of mean value for $R_0$, exploring the full range of values previously observed and reported in literature reviews. Specifically, reported estimates of $R_0$ were similar in the two considered countries, ranging from 1.06 to 3.72 in Haiti and from 1.10 to 3.50 in Cameroon. While eligible priors for $R_0$ were available at the country level in the literature, we did not find country-specific priors for $R_t$. Thus, in our main analysis, we used a Gamma distribution with a mean of 2 and a standard deviation of 0.7 for the prior distribution on $R_t$, consistent with previous research.[50,52] This prior is used in the EpiEstim Bayesian analysis to estimate $R_t$ within a well-informed, limited boundary. In Appendix Figure 3, we explored the local sensitivity of our effective reproduction number estimates to the choice of priors for $R_t$. All sensitivity analyses are available in the Appendix at https://github.com/ehulland/EpiEstim_Cholera/.

In response to rising case and death counts, OCV1 campaigns were conducted in both countries. In Cameroon, an initial campaign was launched in February 2022, followed by a second campaign in April 2022; in Haiti, the vaccination campaign started in mid-December 2022 and continued through January 2023. In Cameroon, the first campaign targeted the South-West region and reached 85.50% administrative coverage, while the second campaign targeted the Littoral, South, and South-West regions and reached 89.00% administrative coverage. In Haiti, the December campaign reached 76.00% administrative coverage on average, with a marked geographical difference (i.e., 69.90% in the Ouest vs.

100.00% in the Centre region).[4,64] In sensitivity analyses, we explored the impact of lower and higher values of the estimated OCV1 coverage on our estimates of vaccine effectiveness.

*Digitizing epidemic curves*

As traditional cholera outbreak surveillance datasets were not readily available, epidemic curves were downloaded from published situation and disease outbreak news reports and digitized using the metaDigitise package in R.[65] Epidemic data for Cameroon were reported weekly. Since the reporting time period (here, seven days) cannot be longer than the SI (here, five days) for estimation purposes, we needed to transform the weekly incidence into daily incidence;[66] we did so using a uniform distribution over the entire seven-day week. For Haiti, case counts were already reported daily and did not require any additional transformation. Of note, these epidemic curve data were the only NMR images converted to a digital format in this study.

*Leveraging EpiEstim and VaxEstim in low-resource settings*

We used the EpiEstim framework and corresponding R package to estimate weekly values of $R_t$, independently for Haiti and for Cameroon.[42,43] The procedure outputs a country-specific posterior distribution of $R_t$ at each time point. Using the mean, 0.025 quantile, and 0.975 quantile from these country-specific Gamma distributions, as well as the corresponding reported administrative OCV1 coverage ($VC_t$) and estimated $R_0$ from the literature, we then employed the following canonical equation[44] to estimate OCV1 effectiveness ($VE_t$) in Haiti and Cameroon over time:

$$VE_t = \frac{1 - \frac{R_t}{R_0}}{VC_t}$$

Notably, we removed estimates outside the lower and upper VE limits of 0% and 100%, respectively. The number and percent of estimates falling outside these bounds is available in Appendix Table 2 (https://github.com/ehulland/EpiEstim_Cholera/).

*Availability of environmental covariates*

In contrast to infectious disease surveillance data, environmental covariates are widely available at high spatial and temporal resolution, consistently across countries. In currently ongoing research, we aim to use environmental covariates and time series modeling (e.g., seasonal ARIMA[67–69]) to identify correlates of higher cholera transmission, as measured by the effective reproduction number. Because cholera has a strong environmental transmission cycle, we now seek to understand the influence of rainfall,[70,71] average air temperature,[72,73] sea surface temperature (SST),[74] and sea surface salinity (SSS),[75,76] on the resurgence and transmission of the disease. Indeed, each of these four environmental covariates has been linked to cholera incidence in prior work and operates indirectly through flooding, pH changes, bacterial concentrations, or algal blooms.[77,78] For our two locations of interest, each of these four covariates— rainfall, air temperature, SST, and SSS—is available daily; this temporal resolution is the same as that of our EpiEstim $R_t$ estimates, allowing us to align the two types of time series – epidemic (outcome) and environmental (exposure) – in our ARIMA models.

*Data availability and role of the funders*

All analyses were done using R version 4.3.1.[79] This research was supported in part by the National Institute of General Medical Sciences, National Institutes of Health (R35GM146974), the National Science Foundation (SES2200228 & IIS2229881), and the Eric & Wendy Schmidt Center of the MIT-Harvard Broad Institute. The funding sources had no involvement in the study design; in the collection, analysis, and interpretation of data; in the writing of the report; or in the decision to submit the paper for publication.

**Results**

In Cameroon, 6,652 confirmed and suspected cases were officially reported between October 10, 2021 and April 24, 2022; in Haiti, 58,230 confirmed and suspected cases were reported between October 1, 2022 and August 5, 2023. In parallel, our digitization of each country's epidemic curves yielded a total of 6,616 cases in Cameroon and 54,927 cases in Haiti, resulting in overall relative error rates of 0.54% and 5.67%, respectively (refer to Figure 1 and Appendix 1 Section 1 for the formula used to calculate the relative error rate).

Such aggregate differences between reported and digitized morbidity are quite small and reasonable for the application of EpiEstim methods, particularly in the absence of machine-readable surveillance data. In addition, we performed a visual inspection by overlaying our digitized epidemic curve with its published counterpart for Haiti, where both the absolute cumulative case count (i.e., the denominator) and the relative error rate were larger than in Cameroon. This inspection (Figure 2) suggested that the case count trajectories were largely similar throughout the considered time period. Given that our primary objective is to explain the variation in disease spread, these overall trajectories are considerably more relevant than exact case numbers.

EpiEstim models yielded estimates of $R_t$ for one-week-long rolling windows over the full outbreak periods to date in Cameroon and Haiti (Figure 3). The median initial $R_t$ at the start of the second outbreak week was estimated to be 1.90 [95% credible interval (CI): 1.14-2.95] in Cameroon and 2.60 [2.42-2.79] in Haiti. Interestingly, while $R_t$ generally remained between 0.8 and 1.2 in Haiti, it was much more variable in Cameroon. Specifically, its trajectory showed a distinct oscillating pattern: $R_t$ rose rapidly to 2.50 [1.88-3.25] in the third week, and then fell steeply to 0.30 [0.15-0.52], followed by one last rise to 1.79 [1.32-2.37] before settling around 1, the threshold above which an outbreak propagates. In both countries, $R_t$ values in subsequent periods hovered around this

threshold. In Cameroon, the overall median $R_t$ was 1.07 [0.52-1.94] and nearly two-thirds (126 out of 195, or 64.62%) of all median $R_t$ estimates were above 1. In contrast, in Haiti, the overall median $R_t$ was 0.97 [0.80-1.38] and only about one-third (105 out of 302, or 34.8%) of median estimates were above 1, indicating less potential for an explosive outbreak. Despite the proportion of days with an $R_t$ over 1 being higher in Cameroon, the outbreak lasted much longer in Haiti (44 vs. 29 weeks) and resulted in an overall larger number of cases (58,230 vs. 6,652).

Using published estimates of $R_0$ and administrative statistics for vaccine coverage, we then estimated time-varying OCV1 effectiveness for Cameroon and Haiti separately, following the initiation of vaccination campaigns in each country (Figure 4). In Cameroon, the median OCV1 effectiveness over the period spanning February 18 to April 30, 2022 was 54.88% [18.94–84.90%]; it ranged from a minimum of 17.37% on March 23, 2022 to a maximum of 90.45% on April 17, 2022. In Haiti, the median OCV1 effectiveness over the period spanning December 19, 2022 to August 5, 2023 was 75.32% [54.00–86.39%]; it ranged from a minimum of 51.26% on May 1, 2023 to a maximum of 96.87% on August 5, 2023.

Median values for OCV1 effectiveness are best understood when contrasted with herd immunity thresholds – the level at which the population is no longer susceptible to disease transmission, thanks to immunity acquired through either immunization or previous infection. To calculate such country-specific thresholds, we used the standard approximation of $1-(1/R_0)$[80] for herd immunity and sourced country-specific $R_0$ values from the literature. We found that median OCV1 effectiveness estimates in Cameroon and Haiti exceeded estimated herd immunity thresholds of 48.72% and 55.95%, respectively. However, the estimated median OCV1 effectiveness in each country dropped below these thresholds at certain times, suggesting periods when sustained transmission occurred. In Cameroon, such periods with a high risk of transmission primarily occurred soon after vaccine delivery, in late February and early-to-mid March. Yet by the second round of the vaccination campaign (April 8-12, 2022), OCV1 effectiveness estimates were consistently above the estimated herd immunity threshold. In contrast, in Haiti, these dips below the herd immunity threshold happened well after OCV1 delivery started, in late April to early May and mid-June, likely allowing further disease propagation. Notably, despite a request by the MSPP for a second round of OCV1 delivery, no reports of a second vaccination campaign have been identified.

## Discussion

To our knowledge, this study is the first to contrast cholera outbreaks in two distinct countries and to assess the relative impact of OCV1 campaigns on local transmission using open-access, non-machine-readable (NMR) outbreak case data and EpiEstim software. Additionally, our work provides the first application of the VaxEstim extension to a low-resource setting. Prior to this study, the VaxEstim extension has been used only in high-income Ohio, USA. Whereas prior work leveraged curated surveillance data from robust electronic health records,[45] ours relies exclusively on NMR case counts published in situation and disease outbreak news reports. The digitization methods we used to extract epidemic curve data from NMR images yielded reasonable estimates of cholera cases over time, with an overall discrepancy between actual and inferred case counts of approximately 5% or less. For many contemporary and historical outbreaks, static images of epidemic curves are the only publicly available data source to track case development in near real time. Indeed, structured, well-formatted surveillance datasets (e.g., line lists) are often highly privatized—if they exist at all. By using data from only situation and disease outbreak news reports, along with parameters extracted from the published literature, we demonstrate that EpiEstim and VaxEstim can be employed quickly during outbreaks occurring in data-scarce settings to aid in decision-making and monitoring. Going forward, we encourage other infectious disease researchers and mathematical modelers to consider such tools when data are not immediately accessible in a machine-readable format.

In both Haiti and Cameroon, we found that outbreaks initially caused large surges in the effective reproduction number, $R_t$. Although the point estimate for $R_t$ in Haiti (2.60 [95% CI: 2.42-2.79]) was higher than that in Cameroon (1.90 [1.14-2.95]), the corresponding credible intervals partially overlapped, owing in part to greater uncertainty in $R_t$ estimates for Cameroon. Notably, our results corroborate several patterns of cholera transmission reported in prior studies. First, the order of magnitude we estimated largely aligns with values of $R_0$ previously reported in the literature, whether specifically in Haiti and Cameroon or in other sub-Saharan countries.[50,52,58–63] While $R_0$ and $R_t$ are markedly different metrics, prior studies have relied on estimates of $R_t$ obtained during the first week of an outbreak to generate an "early outbreak reproductive number".[52] Indeed, such a value can be more closely compared to $R_0$ than later outbreak estimates of $R_t$ because interventions to curb transmission are rarely implemented in the first week following the index case.[52] Second, in both countries, these initial periods — marked by high values of $R_t$ — were followed by longer periods during which $R_t$ oscillated around 1 but was never sufficiently low for transmission to cease completely. This temporal pattern is consistent with prior analyses of $R_t$ dynamics for cholera.[48–51]

Most importantly, our results suggest that OCV1 campaigns in Haiti and Cameroon were generally effective, corroborating field surveys of single-dose vaccine effectiveness conducted in other countries, including South Sudan, Bangladesh, and the Democratic Republic of the Congo.[32–34] Although our point estimates of OCV1 effectiveness were predominantly higher in Haiti than in Cameroon, the corresponding country-specific credible intervals overlapped. The absence of a statistically

significant difference between the two countries suggests that the effectiveness of the vaccine was similar across both contexts, despite differences in population density, ease of implementation, health and water infrastructure, and cholera outbreak history. This key finding indicates that the success of OCV1 campaigns depends less on external factors than do other vaccination campaigns (e.g., cold chain maintenance for certain routine immunizations like the oral polio vaccine[81] or higher dropout rates for Diphtheria-Tetanus-Pertussis three-dose vaccine (DTP3) and several other routine immunizations requiring multiple doses[82,83]). From a global health equity standpoint, such robustness in vaccine effectiveness across differing country settings is encouraging: it signals that the implementation of OCV1 campaigns in response to new outbreaks should similarly work in other locations.

Since single-dose campaigns are a relatively new recommendation, only dating back to October 2022, the literature on OCV1 field studies to date remains sparse. While field-based assessments of ongoing or upcoming OCV1 campaigns will provide additional confirmatory evidence regarding vaccine effectiveness, our work demonstrates a method that complements on-the-ground surveys but requires much less time and funding to conduct. Thus, this approach would be especially relevant in locations without the resources to implement field studies of vaccine effectiveness. In addition, using digitization and EpiEstim early in an outbreak could additionally provide near real-time information about ongoing transmission, while local applications of VaxEstim could provide dynamic insights into vaccine allocation. However, understanding the local context, drivers of disease transmission, and barriers to efficient vaccine delivery also remain critical when considering such allocation.

$R_t$ – the key marker of disease transmission – depends on the contact rate between people, the duration of the infectious disease period, and the probability of infection given contact. Since these factors may differ between Haiti and Cameroon, $R_t$ is a relevant metric to identify the most important mechanisms at play in each location.[84] Notably, Haiti has a much higher population density than Cameroon (415 vs. 58 people per km$^2$, respectively),[85] allowing for much higher rates of contact between individuals, which can in turn lead to explosive initial rates of transmission. The population in Haiti also has significantly lower access to improved water sources than that in Cameroon (68.29% vs. 73.95%), which may increase a person's likelihood of consuming contaminated water (Appendix Table 1: https://github.com/ehulland/EpiEstim_Cholera/).

Previous research has suggested additional factors that can influence vaccine effectiveness, from system-level factors like cold chain effectiveness to individual-level factors like the nutrition status of the individual receiving the vaccine or their previous history of cholera infection.[86–88] Although these factors may explain some of the variability in vaccine effectiveness estimates that we observed between Haiti

and Cameroon, it is important to re-emphasize that their credible intervals overlapped, suggesting relatively similar vaccine effectiveness across both contexts. While such similarity brings great promise for the deployment of OCV1 vaccines in other low-income and conflict settings affected by cholera outbreaks, the aforementioned country-specific factors should still be considered when planning future vaccination campaigns.

Although cholera is much newer to Haiti than it is to Cameroon, the outbreaks that have affected Haiti in the last decade have been much larger and more widespread.[89–92] While previous cholera infection is not fully protective against future infection, current research suggests that patients may acquire some short-term immunity from prior infection.[86] Similarly, frequent exposure to cholera does seem to boost the immune response of people living in endemic countries.[86] In Haiti, where the 2010 outbreak led to over 800,000 cases between 2010 and 2019, we hypothesize that, at the start of the 2022-2023 outbreak, the fraction of the population who was fully susceptible to the disease was smaller than that in Cameroon, where more frequent but smaller outbreaks have historically occurred. [89–92] This assumption is further supported by findings from the most recent outbreak that occurred in Haiti, namely that the largest proportion of reported cases was in children under five who were not born when prior outbreaks occurred.[89] Moreover, we found higher overall estimates of $R_t$ in Cameroon than in Haiti, supporting our hypothesis that the larger baseline proportion of the population who was susceptible contributed to sustained transmission cycles throughout the country's outbreak. In addition to some existing immunity, the severe outbreaks that have ravaged Haiti have helped develop the institutional knowledge necessary to prepare for subsequent large-scale outbreaks, improve health infrastructure, and fulfill vaccine needs more effectively. Such systemic efforts may also correlate with behavioral health improvements. For instance, we believe that greater awareness of the impact of cholera might also prompt higher adoption of existing treatments such as ORT or uptake of preventive interventions like better handwashing and water practices. This hypothesis is supported by recent data on ORT uptake among diarrheal children, estimated at 39.05% [29.17-49.41%] in Haiti vs. 19.91% [11.95-31.05%] in Cameroon.[93] In contrast to Haiti, which has had only two full vaccination campaigns and one pilot program to date,[93,94] Cameroon has had a large number of OCV campaigns since the first one was launched in 2015, particularly within the past 5 years.[94,95] These successive campaigns may have contributed to higher vaccine coverage in Cameroon than in Haiti, despite lower levels of vaccine confidence.[95] Ultimately, this combination of factors resulted in lower vaccine effectiveness in Cameroon than in Haiti, because of its inverse relationship with vaccine coverage.

In contrasting the outbreaks that affected Haiti and Cameroon, we additionally considered environmental drivers of cholera transmission. As previously noted,

environmental data on precipitation, air temperature, sea surface temperature, and sea salinity are openly accessible, in machine-readable format, for the entirety of the globe and on a routine temporal basis. Paired with time series models, these environmental data can provide valuable insights to prepare for future outbreaks in Haiti and Cameroon. To explain and subsequently forecast the spread of cholera as a function of evolving environmental conditions in a given country, these data can also be incorporated into disease transmission models. However, if we were to scale that approach to other countries, we would be limited by the availability and quality of infectious disease surveillance data. Indeed, the richness and wide availability of environmental data is in stark contrast to most infectious disease surveillance data, which are often inaccessible, irregularly available, and critically not standardized across geographical contexts or diseases. Given the rise of emerging infectious diseases and the resurgence of existing ones,[96] a focus on generating high-quality, harmonized infectious disease surveillance data could have profound effects on our ability to model and forecast disease spread around the globe.

*Limitations*

While the digitization of NMR epidemic curves from situation and outbreak reports allows modelers to expand outbreak analyses by leveraging novel data sources rather than relying solely on traditional surveillance data, their use brings some challenges. First, the digitization of epidemic curves, which provided the daily time series which underlie our analyses, can result in estimated daily counts not precisely in line with reported case counts (i.e., a relative error rate of 0.54% and 5.67% in Cameroon and Haiti, respectively). Although the discrepancy between digitized and reported case counts did not significantly impact our case study, outbreak trajectories may not always be well-captured by the digitization process (e.g., low relative case counts may be hard to distinguish from zeros during the digitization process when daily case counts are otherwise high). However, we emphasize that for many outbreaks, the digitization of NMR epidemic curve images can allow for computational analyses where previously no such ability existed. Second, detailed information required for mechanistic compartmental models of cholera transmission is not always openly accessible. Examples include the size of the fully susceptible population in each area, records of chains of transmission and individual symptom onset dates, and assumptions about the underlying disease process and corresponding parameterization. Since phenomenological models like those underlying EpiEstim do not rely on assumptions about the disease process and generally require less parameterization, they are often less precise than mechanistic compartmental models. However, a trade-off exists between the precision of epidemiological models and their practicality and relevance for decision makers. Here, we demonstrated that using EpiEstim allows estimation of $R_t$, even early in an outbreak, without

requiring more information than the SI of a disease and the current epidemic curve. A strong advantage of this built-in R package is its accessibility, allowing practitioners at all levels to make informed, local, and timely decisions.

In addition to using openly accessible outbreak case data, we also relied on published reports of vaccine coverage, which were relatively sparse. We therefore did not have access to the exact location or timing of OCV1 delivery and instead used the overall coverage as reported. Specifically, we assumed a simplified coverage timeline, with instantaneous coverage on the first reported date. Yet, by doing so, we were able to extend the application of VaxEstim to places where vaccine coverage surveys are not conducted or where the resulting data are not openly available. Moreover, in our estimation of vaccine effectiveness, we are combining information at two different geographical resolutions: on the one hand, our vaccine coverage data represent only specific regions of each country, corresponding to locations where campaigns were conducted; on the other hand, outbreak case data are most frequently analyzed at the national level. In both Haiti and Cameroon, however, OCV1 campaigns targeted the most highly affected regions, likely accounting for a substantial portion of all cases. Notably, in our sensitivity analysis (Appendix Figure 6: https://github.com/ehulland/EpiEstim_Cholera/) focused on the Ouest department in Haiti, we found slightly higher vaccine effectiveness than in our main analysis reflecting the whole country; yet the difference was not statistically significant. Similarly, our sensitivity analyses using different levels of $R_0$ (Appendix Figure 5) yielded different point estimates for $R_t$ and vaccine effectiveness, although the corresponding credible intervals again overlapped with those reported in our main analysis. These sensitivity analyses further emphasize the need for rapidly sharing local data and for conducting epidemiological analyses early in an outbreak to facilitate targeted decision-making.

## Acknowledgements

We would like to thank Leslie Gaffney at the MIT-Harvard Broad Institute for her help in reviewing and revising Figure 2.

## Copyright

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

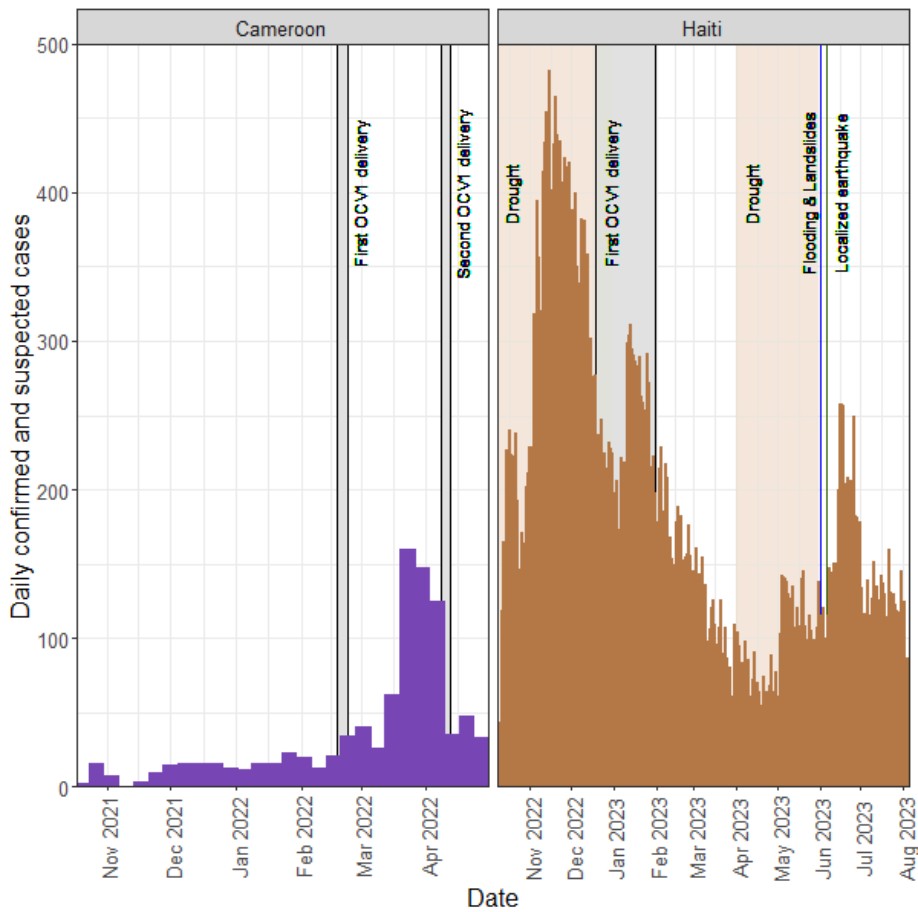

**Figure 1: Epidemic curves for cholera outbreaks.** Epidemic outbreak case data for Cameroon (in purple) and Haiti (in gold) are shown by date (x-axis). The y-axis shows all daily confirmed and suspected cases. For each country, color-coded segments correspond to specific events: OCV1 campaigns (light gray); major rainfall, flooding events, and related landslides (blue),[97] droughts and low rainfall events (light beige),[98] and earthquakes (dark green).[99]

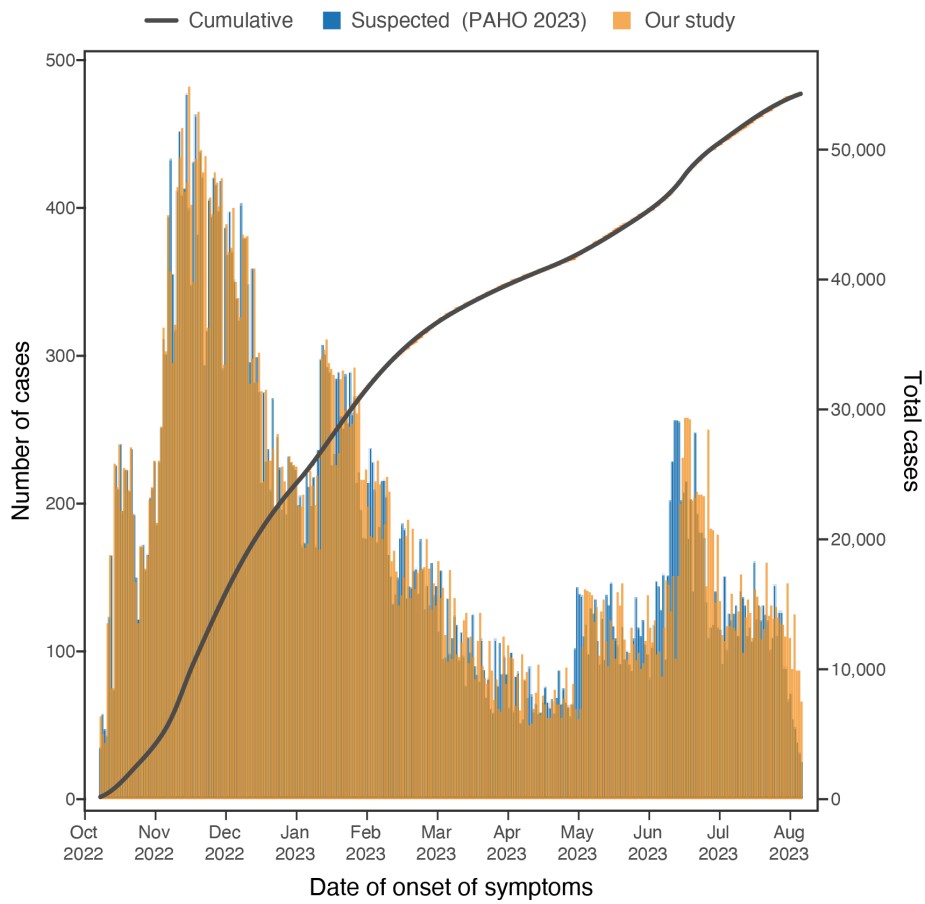

**Figure 2: Reported and digitized epidemic curves, Haiti.** This figure shows the published suspected and confirmed daily outbreak case counts (blue bars) and the overall cumulative case count (black curve), as reported by MSPP and PAHO in 2022-2023[35,38], overlaid with the digitized daily confirmed and suspected case counts (gold bars) obtained using the metaDigitise package in R[65].

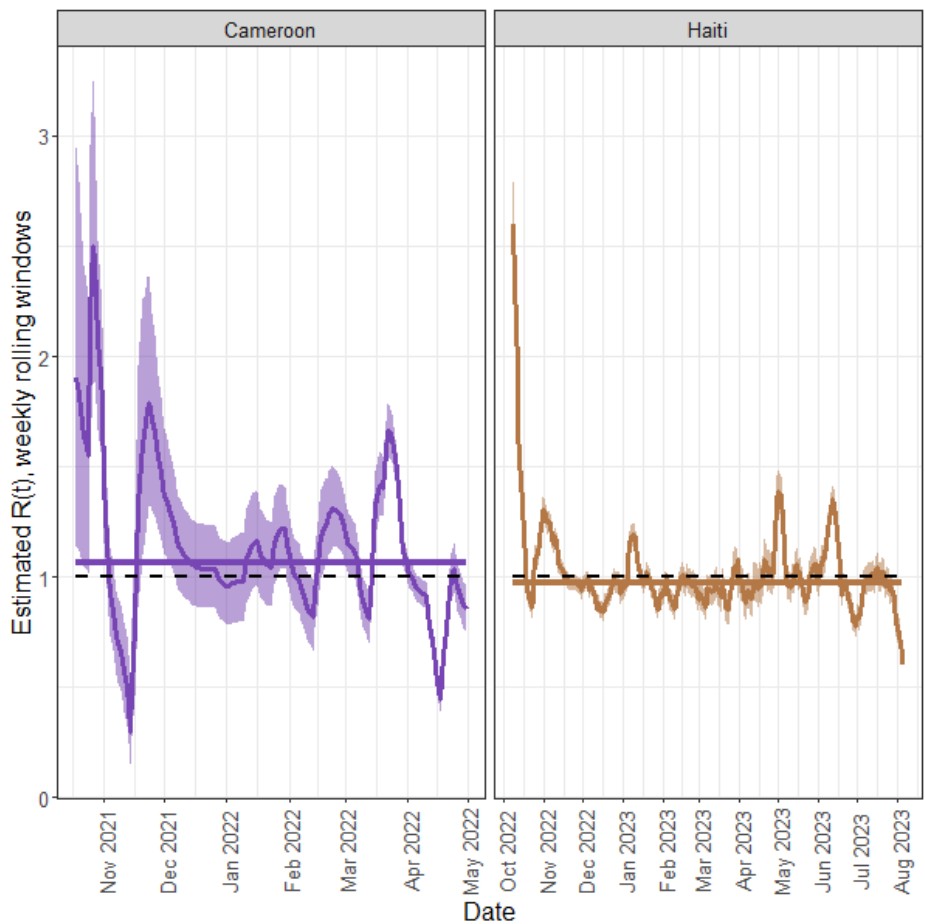

**Figure 3: Estimated $R_t$ over time in Cameroon and Haiti.** This figure depicts the estimated $R_t$ per weekly rolling window over the country-specific study period for Cameroon (purple, left panel) and Haiti (gold, right panel). For each country, the shaded region represents the 95% credible interval and the corresponding solid line represents the overall median $R_t$. On both panels, the dashed black line represents the threshold for disease propagation (i.e., $R_t > 1.0$).

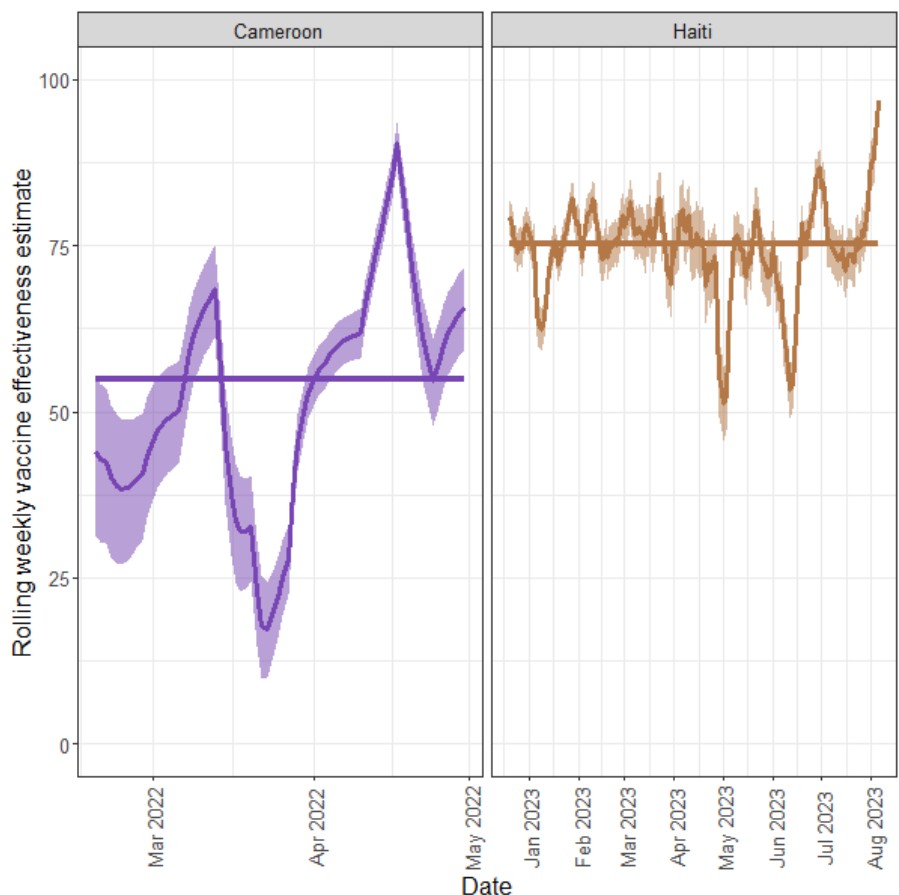

**Figure 4: Estimated rolling weekly vaccine effectiveness over time.** This figure depicts the estimated vaccine effectiveness in weekly rolling windows over the study period for Cameroon (purple, left panel) and Haiti (gold, right panel). For each country, the shaded region represents the 95% credible interval and the corresponding colored solid line represents the overall median vaccine effectiveness. The start date for our estimates corresponds to the first day following the beginning of the vaccination campaign.

