# OpenReview forum: "Estimating time-varying cholera transmission and oral cholera vaccine effectiveness in Haiti and Cameroon, 2021-2023"
_KDD.org/2024/Workshop/epiDAMIK — KDD 2024 Workshop epiDAMIK_

### Official Review · Reviewer_zXHD · 2024-06-26

**Rating:** 3
**Confidence:** 3

**Review:**

Summary:

This paper explores the epidemiology modeling of cholera in Haiti and Cameroon using EpiEstim/VaxEstim. This work focuses particular on the challenge of a low-resource setting, relying exclusively on non-machine-readable (NMR) case counts published in situation and disease outbreak news reports.

The results demonstrate overall relative error rates of total cases in general, with showing the estimated $R_t$ over time, underscoring the value of combining NMR sources of outbreak case data with computational techniques and the utility of VaxEstim for rapid, inexpensive estimation of vaccine effectiveness in data-poor outbreak settings.

Suggestions:

1. Paper presentation should be improve. For example, using latex gives better presentation than current. Also resize the figure to a proper size, such that they do not look like as stretched.

2. The contribution should be more emphasized. For example, in the abstract, the author should also mention the reasonableness of the results, compared to the ground truth or other official literature, such that the meaningfulness of the employed approach in this paper can be recognized.

3. While the overall relative error rate over the whole epidemiology time period is small, the author may include additional discussion on the rolling $R_t$/$R_0$ by the model. It would be more insightful if changes of these values can be interpreted with specific events or interventions, such as any possible case studies.

---

### Official Review · Reviewer_ZkD2 · 2024-06-30
**Study reporting outbreak characteristiccs and vaccine effectiveness in two data-poor settings.**

**Rating:** 3
**Confidence:** 1

**Review:**

This study contrasts cholera outbreaks in Haiti and Cameroon including vaccine effectiveness.
This paper describes how to rapidly estimate outbreak characteristics by applying an open source method, EpiEstim, to sparse data.
Quality: Sound methodology. The results are corroborated by prior studies on Cholera outbreaks.
Clarity: Clearly written.
Originality: The contrastive study of vaccine effectiveness in two countries. Aside from the data-poor setting, the method does not have much novelty.
Significance: for epidemiologists and global health policy-makers.